# Effects of RDL GABA Receptor Point Mutants on Susceptibility to Meta-Diamide and Isoxazoline Insecticides in *Drosophila melanogaster*

**DOI:** 10.3390/insects15050334

**Published:** 2024-05-06

**Authors:** Tianhao Zhou, Weiping Wu, Suhan Ma, Jie Chen, Jia Huang, Xiaomu Qiao

**Affiliations:** 1Ministry of Agriculture Key Laboratory of Molecular Biology of Crop Pathogens and Insects, Institute of Insect Sciences, Zhejiang University, Hangzhou 310058, China; 12116111@zju.edu.cn (T.Z.); 3170100432@zju.edu.cn (W.W.); 22116097@zju.edu.cn (S.M.); huangj@zju.edu.cn (J.H.); 2Collaborative Innovation Center of Green Pesticide, National Joint Engineering Laboratory of Biopesticide Preparation, Key Laboratory of Subtropical Silviculture, College of Forestry and Biotechnology, Zhejiang A & F University, Hangzhou 311300, China; chenjie@zafu.edu.cn; 3Xianghu Laboratory, Hangzhou 311231, China

**Keywords:** GABA receptor, meta-diamides, isoxazolines, resistance, point-mutation, CRISPR, *Drosophila*

## Abstract

**Simple Summary:**

Insects have special receptors called RDL that regulate the effects of a neurotransmitter GABA in their nervous system. Interestingly, these receptors are also the main targets of many insecticides used to control pests. We used fruit flies with different variants of the *Rdl* gene and tested four new types of meta-diamide and isoxazoline insecticides. The results showed that some mutations made the flies less sensitive to certain insecticides, while another mutation made the flies super resistant to all of the insecticides tested. Using computer simulations, we explored how these mutations might be affecting the insecticides’ ability to attach. We found that specific areas between parts of the receptor seem to be important for these new insecticides to work. This helps us understand how insects can resist these insecticides and might lead to better ways to control pests in the future.

**Abstract:**

Ionotropic γ-aminobutyric acid (GABA) receptors in insects, specifically those composed of the RDL (resistant to dieldrin) subunit, serve as important targets for commonly used synthetic insecticides. These insecticides belong to various chemical classes, such as phenylpyrazoles, cyclodienes, meta-diamides, and isoxazolines, with the latter two potentially binding to the transmembrane inter-subunit pocket. However, the specific amino acid residues that contribute to the high sensitivity of insect RDL receptors to these novel insecticides remain elusive. In this study, we investigated the susceptibility of seven distinct *Drosophila melanogaster Rdl* point mutants against four meta-diamide and isoxazoline insecticides: isocycloseram, fluxametamide, fluralaner, and broflanilide. Our findings indicate that, despite exhibiting increased sensitivity to fluralaner in vitro, the *Rdl^I276C^* mutant showed resistance to isocycloseram and fluxametamide. Similarly, the double-points mutant *Rdl^I276F+G279S^* also showed decreased sensitivity to the tested isoxazolines. On the other hand, the *Rdl^G335M^* mutant displayed high levels of resistance to all tested insecticides. Molecular modeling and docking simulations further supported these findings, highlighting similar binding poses for these insecticides. In summary, our research provides robust in vivo evidence supporting the idea that the inter-subunit amino acids within transmembrane M1 and M3 domains form the binding site crucial for meta-diamide and isoxazoline insecticide interactions. This study highlights the complex interplay between mutations and insecticide susceptibility, paving the way for more targeted pest control strategies.

## 1. Introduction

γ-aminobutyric acid (GABA) is the major inhibitory neurotransmitter in the insect central nervous system. One key protein involved in this process is the ionotropic GABA receptor, a chloride channel composed of RDL (resistant to dieldrin) subunits [1,2]. Unlike their mammalian counterparts, insect GABA receptors lack the variety of subunits that form diverse hetero-oligomeric structures. In fact, the RDL subunit can form functional homo-oligomeric ion channels, mimicking the pharmacology of native insect receptors [3]. The RDL GABA receptor is an important target for insecticides, including conventional noncompetitive antagonists (NCAs) such as cyclodienes and phenylpyrazoles [4,5]. However, resistance-conferring mutations in the *Rdl* locus have emerged in various insect species, including agricultural pests that threaten food security and mosquitoes that transmit deadly diseases, posing significant challenges for agriculture and public health [1,6,7]. The pivotal mutation typically involves a single amino acid substitution at position 301 (*Drosophila melanogaster* numbering) or the 2′ site based on the index number for the M2 membrane-spanning region. This substitution commonly switches an alanine to a serine [3], but replacements with glycine or asparagine have also been documented. Mutations in these pore-lining residues cause conformational changes that disrupt the binding of NCA insecticides into the channel pore. This reduced binding capability translates to lower insecticide potency and, ultimately, resistance in insect populations.

Meta-diamides (e.g., broflanilide and cyproflanilide) and isoxazolines (e.g., isocycloseram, fluxametamide, and fluralaner) are two novel generations of insecticides acting on RDL GABA receptor. They work by allosterically inhibiting the channel, leading to hyperexcitation and convulsions [8,9,10,11]. Due to their distinct target site on the RDL GABA receptor compared to NCAs, meta-diamides and isoxazolines exhibit no cross-resistance, making them valuable tools for resistance management. Thus, the Insecticide Resistance Action Committee (IRAC) has categorized these insecticides as GABA-gated chloride channel allosteric modulators in the newly assigned mode of action Group 30, distinguishing them from NCAs classified in Group 2 as GABA-gated chloride channel blockers [12].

Recent studies have explored the effects of specific mutations in the RDL subunit on the inhibitory potency of meta-diamides and isoxazolines, offering valuable insights into binding pocket and potential resistance mechanisms [13,14,15,16]. Despite working with different insect species, these studies reveal that RDL receptors are evolutionarily conserved and exhibit similar pharmacology across them. Notably, the heterogeneous expression of RDL with point mutations in M1 and M3 demonstrated significant changes in the in vitro potency of these insecticides (Table 1). For example, I276F and L280C mutations reduced the sensitivity of *Drosophila* RDL receptors approximately sixfold to desmethyl-broflanilide, the active metabolite of broflanilide [8]. By contrast, L280C, but not I276F, substitution showed reduced sensitivity to fluralaner in the housefly (*Musca domestic*) RDL GABA receptor. Interestingly, the I276C mutation enhanced the fluralaner potency 56-fold. For the G335M mutation in the M3 domain, all published results showed significantly reduced potency against meta-diamides and isoxazolines [13,14,15]. In addition, the V339N substitution slightly increased the inhibitory activities of desmethyl-broflanilide. While these in vitro studies indicated the involvement of these amino acid residues and/or nearby residues in the interactions with meta-diamide and isoxazoline insecticides, in vivo genetic functional validation has not been tested, except G335M in a recent report [17].

*Drosophila melanogaster*, owing to its genetic and physiological similarities to other insects, serves as a robust model system in the study of insect toxicology. The application of *Drosophila* genetics significantly enhances our understanding of the mode of action and resistance mechanisms of insecticides [18,19,20]. Here, we generated seven *Drosophila Rdl* mutants with point mutations in the M1 and M3 regions, respectively. Then, we tested the effects of four meta-diamide and isoxazoline insecticides on these mutants. Moreover, we also used molecular modeling to investigate the binding modes and mechanistic effects of these novel insecticides.

## 2. Materials and Methods

### 2.1. Chemicals

Broflanilide (95%TC), isocycloseram (95%TC), fluxametamide (95%TC), and fluralaner (95%TC) were gifts from Dr. Lixin Zhang (Shenyang University of Chemical Technology).

### 2.2. Fly Strains

Flies were maintained and reared on conventional cornmeal agar molasses medium at 25 ± 1 °C and 60% ± 10% humidity with a photoperiod of 12 h:12 h light/dark cycle. The light started at 07:00. The *w^1118^* (#5905) strain was used as the wild-type control in this study. We obtained the *vas-Cas9* (#51324) from the Bloomington Stock Center (Indiana University). The *w[*];mir-1000/TM3, Sb, Ser, twi-GAL4, UAS-eGFP* (#58882) was a gift from Xiaojun Xie lab at Zhejiang University. The following *Rdl* strains were generated in our previous study [21]: *Rdl^3xP3RFP^*, *Rdl^I276F^*, *Rdl^G279S^*, *Rdl^I276F+G279S^*, *Rdl^V339I^*, *Rdl^A343T^*, and *Rdl^A2′A^*.

### 2.3. Generation of Knock-In Flies

To generate the *Rdl* knock-in lines, we utilized a two-step CRISPR (clustered regularly interspaced short palindromic repeats)-Cas9 method by homology-directed repair (HDR) [21] (Appendix A). We designed single-guide (sg)RNAs by E-CRISP (http://www.e-crisp.org/E-CRISP/, accessed on 1 May 2024). The sgRNAs were synthesized and subcloned into the PCFD5 vector by GenScript for expression (Appendix A).

We replaced the attP-3xP3-RFP-loxP sequence in the *Rdl^3xP3RFP^* strain with each of the point mutation alleles in *Rdl* through homologous recombination. We verified candidate 3xP3-RFP-positive or -negative flies by PCR of the targeted region followed by Sanger sequencing. This approach helped us achieve our goals, generating the *Rdl^I276C^* and *Rdl^G335M^* strains. To generate these knock-in lines, we injected a plasmid mixture containing the donor vector and sgRNAs into *vas-Cas9* (#51324). We crossed the *vas-Cas9* (#51324) strain, which carries the 3xP3-GFP marker on chromosome 3, with *Rdl^3xP3RFP^* to produce heterozygous flies as G_0s_ for embryo injections. Typically, we injected 250–300 embryos (UniHuaii) and screened RFP-positive flies under a fluorescence microscope. We then crossed these flies with double-balanced flies (*TM3/TM6B*). The G_1s_ were screened for RFP- and GFP-negative flies, which were crossed with double-balanced flies. Finally, we identified G_2s_ progeny by genomic DNA sequencing, using specific primers designed according to the reported sequences of *Drosophila Rdl* (Appendix A).

### 2.4. Insecticide Bioassays

To evaluate the resistance of different fly strains to insecticides, adult female flies aged between three and five days and of uniform size were used in bioassays. The IRAC susceptibility test method 026 (https://irac-online.org/methods/, accessed on 1 May 2024) was used with minor modifications. In brief, serial dilutions of insecticide solution were prepared in 50 g/L sucrose. Each concentration required approximately 5 mL of insecticide solution. A piece of dental wick (1 cm) was placed in a standard *Drosophila* vial and treated with 800 μL of 5% sucrose with or without insecticide. Ten flies of each genotype were transferred into vials, and each genotype was repeated at least 3 times for every tested insecticide. The vials were kept upside down until all flies became active to avoid flies becoming trapped in the dental wick. After 48 h, the bioassay was assessed, with only the number of dead flies being recorded and omitting seriously affected flies. The LC_50_ resistance ratio (RR) for each *Rdl* mutant was calculated by dividing the LC_50_ value of the mutant by the average LC_50_ of both the *w^1118^* wild-type and the engineered control flies (+/+). This approach helps to account for any potential baseline differences in susceptibility between different genetic backgrounds.

The bioassay of homozygous *Rdl^G335M^* mutant flies was finished three days after ovipositing. Firstly, the adults were transferred into new food-containing tubes to oviposit for 12–16 h after emergence. After 24 h, we collected the larvae through a sieve when they had been soaked in 20% sucrose dilutions for 20 min. Finally, we picked ten homozygous mutants without *GFP* as one replicate by the stereomicroscope and put them individually in single wells of a 48-well plate with 150 μg standard fly food. The survival rate of the flies as recorded on the third day. We performed three replicates for each insecticide and used the *w^1118^* and *Rdl^A2′A^* strain as the control. The probit analysis, using Polo Plus software version 2.0 (LeOra Software, Berkeley, CA, USA), was used for calculating the LC_50_ values and nonlinear log dose–response curves were generated by GraphPad Prism 9.0 (GraphPad Software Inc., La Jolla, CA, USA).

### 2.5. Molecular Docking

To construct a homopentameric model of the *Drosophila* RDL receptor, we used the structural templates derived from human GABA_A_ receptors, specifically the crystal structures bound with picrotoxin and propofol (PDB: 6HUG and 6X3T) [21]. We utilized Molecular Operating Environments (MOEs) to generate the RDL model and perform molecular docking as described previously [22,23]. The docking process was executed utilizing MOE’s Dock application program, with the default parameters in place. Furthermore, 3D protonation was applied, concurrent with the deletion of water molecules, and the energy minimization of both the models and ligands was conducted to optimize their structural stability. The molecular structures of desmethyl-broflanilide and isocycloseram were constructed using MOE builder. Then, we docked these two molecules into the propofol binding site of our RDL receptor model. This served as our initial docking position and the lowest binding energy structure was selected for final analysis.

## 3. Results

### 3.1. Generation and Confirmation of Rdl Mutants

We employed identical CRISPR–Cas9-mediated homology-directed repair (HDR) protocols to produce all *Rdl* mutants. The guide RNA (gRNA) design was reported and all mutants were confirmed by DNA sequencing (Appendix A). In addition to the six *Rdl* strains described in our previous research, we successfully generated two new mutants: *Rdl^I276C^* and *Rdl^G335M^*. Unfortunately, the G335M mutation caused homozygous lethality in adult flies. Therefore, bioassays for this mutant were conducted using homozygous larvae, while all other strains were tested with adult females. To account for potential pleiotropic or off-target effects arising from the two rounds of CRISPR-Cas9 editing, the *Rdl^A2′A^* (+/+) strain was used as the engineered control strain.

### 3.2. Differential Susceptibility of Rdl Mutants to Meta-Diamides and Isoxazolines

We examined the susceptibility of *Rdl* mutant flies carrying different amino acid substitutions against broflanilide, isocycloseram, fluxametamide, and fluralaner (Appendix A; Table 2). Adult bioassays with broflanilide revealed no significant differences in susceptibility among *Rdl* alleles, with LC_50_ values comparable to both wild-type and engineering control files (Appendix A; Table 2).

Our results further revealed varying levels of resistance among tested *Rdl* mutants towards isoxazolines. Compared to control flies, all mutants showed statistically significant differences in their sensitivity to isocycloseram, as evidenced by non-overlapping 95% confidence intervals (Appendix A; Table 2). In particular, the I276C mutation conferred a moderate level of resistance to isocycloseram (RR = 5.7). For fluxametamide, only the I276C mutation (RR = 12.6) and the double mutation I276F+G279S (RR = 4.5) resulted in substantial resistance, while other strains remained largely unaffected (Appendix A; Table 2). Similar to broflanilide, no mutant displayed high levels of resistance to fluralaner (0.43–2.13 in terms of RR). The G279S mutation unexpectedly exhibited a twofold increased sensitivity to fluralaner compared to both wild-type and engineering control flies. However, the double mutant *Rdl^I276F+G279S^* showed a slight but significant increase in resistance (RR = 2.13) to this insecticide (Appendix A; Table 2).

Larval bioassays demonstrated remarkable cross-resistance in the *Rdl^G335M^* mutant to broflanilide, isocycloseram, fluxametamide, and fluralaner. In fact, larval mortality remained below 50% even at the highest concentration (1000 mg/kg) for all insecticides (Table 3). In contrast, flies heterozygous for the G335M mutation showed the same sensitivity to the four insecticides as the wild-type controls when tested at 1 and 10 mg/L. This exceptionally high resistance level underscores the pivotal role of the Gly335 residue in mediating the interaction between RDL receptor and these insecticides within a living organism.

### 3.3. Interactions of Insecticides with RDL Homology Models

To understand how *Drosophila*’s RDL receptor interacts with insecticides, we built a homology model based on the closed-state structure of the human GABA_A_ receptor. This model allowed us to perform docking simulations, predicting how two insecticides, desmethyl-broflanilide and isocycloseram, bind within the allosteric modulator site located at the interface of transmembrane subunits (TSIs). The top-ranked poses from these simulations are shown in Figure 1, highlighting the mutated amino acid residues and their positions within the binding pocket. Interestingly, despite being chemically distinct (meta-diamides vs. isoxazolines), both insecticides adopted remarkably similar horizontal docking positions within the TSI. The Ile276 in M1 and Gly335 in M3, but not the other three residues, appeared to lie in the same plane as the docked insecticides, suggesting their potential roles in ligand interaction and selectivity.

## 4. Discussion

Meta-diamides and isoxazolines are attracting significant attention in the field of pest control, offering a promising alternative to conventional insecticides due to their novel mode of action and favorable toxicological profile. These novel RDL receptor allosteric modulators bind at a distinct site compared to NCAs [11,12,13,14,15,24,25,26,27]. Previous in vitro studies (Table 1) suggested that the M3-M1 interfaces between two adjacent subunits is their likely binding pocket. In this study, we employed a reverse genetics approach to validate this hypothesis by introducing mutations in the *Rdl* gene of *D. melanogaster* using CRISPR/Cas9 genome editing. We generated fly strains with single or double mutations and performed toxicity bioassays against four commercial insecticides: broflanilide, isocycloseram, fluxametamide, and fluralaner.

Our results provide direct in vivo confirmation that the G335M mutation confers high resistance to all four tested insecticides (Table 3). Moreover, our homology modeling and molecular docking simulations indicated that both desmethyl-broflanilide and isocycloseram exhibit similar ligand-binding poses and conformations, with Gly355 positioned in close proximity to both ligands. This aligns with previous in vitro studies [13,14] and a recent investigation that created a *D. melanogaster Rdl^G335M^* strain via a different method, demonstrating high resistance to broflanilide and fluralaner [17]. Consistent with the findings reported in that study, we also found that *Rdl^G335M^* mutants failed to develop into adults. This suggests that Gly335 plays a crucial role in insect survival, making it unlikely for resistance mutations at this specific residue to evolve naturally in the field.

On the contrary, the effects of mutations at four other sites on insecticide susceptibility are not uniform for each mutation/insecticide combination compared to published pharmacological data (Table 1 and Table 2). For instance, an electrophysiological study reported a 56-fold increase in fluralaner sensitivity for the *M. domestica* I274C (equivalent to I276C in *Drosophila*) substitution in the RDL receptor [14]. However, the *Drosophila* strain carrying the I276C mutation did not show enhanced sensitivity to fluralaner. Instead, these flies exhibited medium levels of resistance to two other isoxazolines: isocycloseram and fluxametamide. Moreover, while in vitro studies suggest that the I276F mutation renders RDL less sensitive to desmethyl-broflanilide [13], our *Rdl^I276F^* mutants exhibited normal broflanilide sensitivity. Interestingly, the combination of Gly279 and Ile276 mutations appears to affect isoxazoline binding to the TSI domain. The *Rdl^I276F+G279S^* mutants exhibited statistically significant resistance to all three tested isoxazolines, suggesting the potential cooperative effects of these mutations (Figure 1B; Table 2).

The observed discrepancy between the in vitro and in vivo results in this study underscores a complex issue in insecticide toxicology research. Previous in vitro studies have typically involved expressing recombinant RDL in cell cultures, offering a controlled environment to analyze specific molecular interactions [13,14,15,16]. However, such a simplified system lacks the complex biological processes and feedback mechanisms present in a whole organism. In addition, mutations may trigger compensatory changes elsewhere in the organism’s genome, and indirect impacts on metabolism, detoxification pathways, or other physiological processes may influence insecticide susceptibility in ways not captured by isolated in vitro studies [28,29,30,31,32]. Therefore, integrating and interpreting data from both in vitro and in vivo studies is critical to gain a comprehensive understanding of how mutations influence insecticide susceptibility in real-world scenarios. Given the yet-to-be-elucidated structure and assembly of native insect GABA receptors, genetic evidence may play a pivotal role in advancing this field of research.

The evolution of resistance in insects to commonly used insecticides has become a significant challenge in agricultural and public health contexts. Understanding the mechanisms underlying this resistance is crucial for the development of effective pest management strategies. In this study, we leverage the *Drosophila* model organism to investigate the mode of actions of two promising insecticide classes: meta-diamides and isoxazolines. Our findings contribute to a deeper understanding of how these insecticides interact with their targets, offering valuable insights into the potential mechanisms for future resistance management strategies. Additionally, our research outcomes have the potential to inform and guide the design and refinement of innovative insecticides. Such advancements hold significant relevance for the ongoing management of agricultural and sanitary pest populations, ensuring the sustainability of pest control strategies. A recent study showed that the N316L mutation in the M2 domain renders housefly RDL almost insensitive to fluralaner; further genetic study should test mutations near this region. During the finalization of this manuscript, a separate study produced G335A, I276C, and I276F knock-in *Drosophila* mutants^20^. Interestingly, in their study, homozygosity for I276C was found to be lethal, a result that differs from our findings. Moreover, homozygous G335A flies reached the adult stage and exhibited a high resistance to broflanilide and isocycloseram, indicating that larger amino acid residues at position 355 impose stronger fitness costs.

## Figures and Tables

**Figure 1 insects-15-00334-f001:**
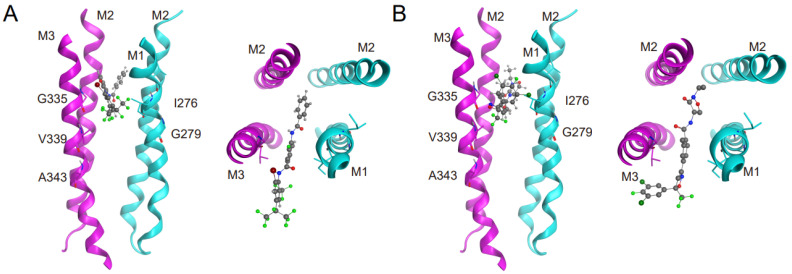
Docking prediction of desmethyl-broflanilide and isocycloseram with *Drosophila* RDL receptor homology models. (**A**) Side view (**left**) and top view (**right**) of the TSI formed by two adjacent subunits with desmethyl-broflanilide (carbon atoms in gray, nitrogen atoms in blue, oxygen atom in red, sulfur atom in yellow, halogen atoms in green). (**B**) Side view (**left**) and top view (**right**) of the TSI formed by two adjacent subunits with isocycloseram.

**Table 1 insects-15-00334-t001:** Changes in potency of *D. melanogaster* and *M. domestic* RDL receptor point mutations to meta-diamides and isoxazolines, as assessed by cell-based in vitro assay. Amino acid positions of mutations are numbered based on the *Drosophila* RDL protein (GenBank accession number: AAF50311). Broflanilide is a pro-insecticide; thus, the active metabolite desmethyl-broflanilide (DMBF) was used.

Region	Point-Mutation	Insecticides	Potency Changes	Reference
M1	I276F	Broflanilide	5.7	[13]
		Fluralaner	0.7	[14]
	I276C	Fluralaner	0.06	[14]
	L280C	Broflanilide	5.7	[13]
		Fluralaner	23.0	[14]
M3	G335M	Broflanilide	>344.8	[13]
		Isocycloseram	50.3	[15]
		Fluralaner	>116.8	[14]
	V339N	Broflanilide	0.13	[16]

**Table 2 insects-15-00334-t002:** Sensitivity levels of the six *Drosophila* mutants and two control lines to four different insecticides. LC_50_ was calculated using log (agonist) versus response nonlinear fit, n = 3–4 trials, 3 replicates per trial.

Insecticides	Strain	LC_50_ (mg/L)	95% CL	Resistance Ratio
Broflanilide	*w^1118^*	0.14	0.10–0.20	1
	+/+	0.14	0.13–0.17	
	I276F	0.16	0.14–0.19	1.14
	I276C	0.25	0.19–0.31	1.79
	G279S	0.10	0.06–0.12	0.71
	I276F+G279S	0.15	0.11–0.33	1.07
	V339I	0.13	0.12–0.15	0.93
	A343T	0.13	0.11–0.14	0.93
Isocycloseram	*w^1118^*	8.0	6.9–9.1	1
	+/+	9.7	9.2–10.1	
	I276F	13.9	11.3–17.1	1.6
	I276C	49.3	39.0–62.7	5.7
	G279S	22.5	18.5–29.7	2.6
	I276F+G279S	15.3	11.0–25.1	1.8
	V339I	16.9	12.1–26.2	1.9
	A343T	18.6	16.2–21.2	2.1
Fluxametamide	*w^1118^*	2.3	1.7–2.9	1
	+/+	5.6	4.5–6.9	
	I276F	4.9	3.4–10.0	1.5
	I276C	41.0	28.4–55.3	12.6
	G279S	2.5	1.7–3.6	0.8
	I276F+G279S	14.7	11.7–18.4	4.5
	V339I	5.0	4.2–6.1	1.6
	A343T	5.9	5.0–7.4	1.9
Fluralaner	*w^1118^*	0.23	0.18–0.28	1
	+/+	0.23	0.18–0.28	
	I276F	0.21	0.17–0.25	0.91
	I276C	0.33	0.22–0.49	1.43
	G279S	0.10	0.09–0.11	0.43
	I276F+G279S	0.49	0.37–0.63	2.13
	V339I	0.18	0.15–0.21	0.78
	A343T	0.20	0.18–0.22	0.87

**Table 3 insects-15-00334-t003:** Sensitivity levels of the homozygous G335M larvae and two controls to four different insecticides. LC_50_ was calculated using log (agonist) versus response nonlinear fit, n = 3–4 trials, 3 replicates per trial.

Insecticides	Strain	LC_50_ (mg/kg)	95% CL	Resistance Ratio
Broflanilide	*w^1118^*	0.8	0.5–1	1
	+/+	0.6	0.5–0.9	
	G335M	>1000	/	>1428
Isocycloseram	*w^1118^*	2.5	2.0–3.8	1
	+/+	2.8	2.1–6.6	
	G335M	>1000	/	>377
Fluxametamide	*w^1118^*	2.5	1.9–3.3	1
	+/+	2.7	1.8–4.5	
	G335M	>1000	/	>384
Fluralaner	*w^1118^*	2.9	2.2–3.6	1
	+/+	2.8	2.0–5.8	
	G335M	>1000	/	>350

## Data Availability

All data are contained within the manuscript and the Appendix A.

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
