# Peer review of "Effects of RDL GABA Receptor Point Mutants on Susceptibility to Meta-Diamide and Isoxazoline Insecticides in Drosophila melanogaster"

_insects, 2024, doi:10.3390/insects15050334_

Round 1

Reviewer 1 Report

Comments and Suggestions for Authors

Zhou et al. engineered transgenic Drosophila lines carrying different variants of RDL, a subunit forming homopentameric GABA-gated-chloride channels. These channels are targeted by several insecticides belonging to different chemical classes such as fiproles, cyclodienes, meta-diamides and isoxazolines. The authors investigated members of the latter two classes such as broflanilide and isocycloseram against different transgenic fly lines expressing mutant RDL receptors. Furthermore, the authors provided modeling and docking studies to support their findings made in in vivo fly bioassays with the different mutant lines. One of the generated RDL mutations, G335M, was lethal in homozygous adults.

The paper is well-written, albeit lightly referenced. It is concise regarding the presentation of the generated results and their discussion.

However, my main concern is based on the fact that only very recently (as pointed out by the authors in the discussion) Ozoe et al. (2024, Ref. 20) published a paper highlighting a similar approach, i.e., investigating the impact of RDL mutations in transgenic flies exposed to broflanilide and isocycloseram. These authors introduced in transgenic flies the same RDL mutations as reported here, e.g., I276F and I276C. Therefore, I must admit that the paper is in many places similar to the Ozoe et al. (2024) paper. Accordingly, the findings presented by Zhou et al. are rather incremental and do not substantially advance the field.

Some additional points are listed below.

11)      Line 15: replace versions by variants.

22)      Line 21: What is meant by “killers”?

33)      Line 71: Insert reference for the IRAC classification.

44)  Line 90: The paper cited (ref. 14) is largely build on previous papers published by Scott & Buchon (Pesticide Biochemistry and Physiology 161 (2019) 95–103), and Douris et al. (Pesticide Biochemistry and Physiology 167 (2020) 104595).

55)  Line 138: Formulated insecticides? The M&M section only lists active ingredients, no formulations.

66)      Line 183-186: Move to M&M section.

77)      Figure 1 and Table 2 are redundant. Move Fig. 1 to the supplement.

88)      Line 241: Insert reference after “profile”.

99)      Line 242: Insert reference after “NCAs”.

110)   Line 251: Add data for heterozygotes to the manuscript.

111)   Line 267: Nakao et al. (2013) did not report desmethyl-broflanilide insensitivity, but 6-fold decrease in inhibitory activity. EC50´s were still in the nanomolar range.

112)   Line 286: Replace development by evolution.

Comments on the Quality of English Language

Minor misspellings need correction

Author Response

1) Line 15: replace versions by variants.

Done.

2) Line 21: What is meant by “killers”?

Revised to “insecticides”.

3) Line 71: Insert reference for the IRAC classification.

We have added reference.

4) Line 90: The paper cited (ref. 14) is largely build on previous papers published by Scott & Buchon (Pesticide Biochemistry and Physiology 161 (2019) 95–103), and Douris et al. (Pesticide Biochemistry and Physiology 167 (2020) 104595).

We have added these references too.

5) Line 138: Formulated insecticides? The M&M section only lists active ingredients, no formulations.

Sorry for the mistake and we used pure chemicals as described.

6) Line 183-186: Move to M&M section.

Moved to M&M section, Line 146-150.

7) Figure 1 and Table 2 are redundant. Move Fig. 1 to the supplement.

We have moved Fig.1 to the Supplement.

8) Line 241: Insert reference after “profile”.

All relevant references were added together in Line 242.

9) Line 242: Insert reference after “NCAs”.

Added.

10) Line 251: Add data for heterozygotes to the manuscript.

We have tested the heterozygous adults in bioassays and there is no resistance. We have added this description in Line 207.

11) Line 267: Nakao et al. (2013) did not report desmethyl-broflanilide insensitivity, but 6-fold decrease in inhibitory activity. EC50´s were still in the nanomolar range.

We have revised it to “less sensitive”

12) Line 286: Replace development by evolution.

Done.

Reviewer 2 Report

Comments and Suggestions for Authors

This research adopted the reverse genetic experiments to construct the mutant    fruit fly strains to reveal the key molecular targets of meta-diamide and isoxazoline insecticides in Drosophila melanogaster.  The results provided potential basis for predicting the resistance mechanism of these insecticides when they will be widely  applied to control agricultural pests in the future. There are 2 points need the authors to explain.

1 There are not significant resistance of broflanilide observed between the 6 mutant strains and the wild strain, is it because the mutant did not confer the resistance of broniflanilide? Have the authors  ever tested the resistance of these mutant strains to desmethy- broflanilide? Is it possible the mutant fly resistant to desmethy- broflanilide?

2 Besides transmembrane M1 and M3 domains, there  also might be key target site in M2 domain which is important for isoxazoline insecticides binding, it is suggested to discuss it too ( pharmacological experiments showed that housefly RDL GABAR N316L mutant was almost insensitive to fluralaner).

Author Response

1 There are not significant resistance of broflanilide observed between the 6 mutant strains and the wild strain, is it because the mutant did not confer the resistance of broniflanilide? Have the authors ever tested the resistance of these mutant strains to desmethy- broflanilide? Is it possible the mutant fly resistant to desmethy- broflanilide?

Broflanilide is a prodrug and is metabolized to desmethyl-broflanilide when ingested in insects. It is basically same to feed broflanilide or its metabolites in fly bioassays.

2 Besides transmembrane M1 and M3 domains, there  also might be key target site in M2 domain which is important for isoxazoline insecticides binding, it is suggested to discuss it too ( pharmacological experiments showed that housefly RDL GABAR N316L mutant was almost insensitive to fluralaner).

We have added the following discussion in Line 296: "A recent study showed that the N316L mutation in the M2 domain render housefly RDL almost insensitive to fluralaner, further genetic study should test mutations near this region”.

Reviewer 3 Report

Comments and Suggestions for Authors

Zhou et al. generated several mutant RDL receptors in the Drosophila line to test resistance levels in an in vivo system. The results provided information on key mutants related to different insecticides. I have several comments that the authors need to address.

Major comments:

1.     It is helpful to have a table in the introduction describing previous studies. However, some studies were based on different insects, such as the house fly and cotton leafworm. The authors should include the species in the table, as well as the similarities and differences between those species' RDL receptors and Drosophila, either in the introduction or discussion.

2.     It is interesting that G335M larvae were unsuccessfully developed into adults. Since the authors tested G335M resistance in the larval stage, they should have also included other mutant strains examined in the larval stage to allow for better comparison, at least those with major differences in the adult stages (I276C, I276F+G279S, G279S).

3.     The discussion is incomplete. The authors mentioned different results about I276C from published paper. Moreover, same study successfully generated G335A into the adult stage where G335M failed. The authors should have discussed the differences and potential reasons for the discrepancies between the two studies.

Minor comments:

1.     The citations need improvement. For example:

Line 52-55: Where is the citation about RDL in mosquitoes and pests?

Line 72-74: Citation of the recent study?

Line 276-278: Reference?

Line 278-281: Where do the references support this conclusion?

2.     Line 118: The format of the link is different.

Author Response

  1. It is helpful to have a table in the introduction describing previous studies. However, some studies were based on different insects, such as the house fly and cotton leafworm. The authors should include the species in the table, as well as the similarities and differences between those species' RDL receptors and Drosophila, either in the introduction or discussion.

We have added the species in the Table 1 title and the following sentence in the Line 74: “Despite working with different insect species, these studies reveal that RDL receptors are evolutionarily conserved and exhibit similar pharmacology across them”.

  1. It is interesting that G335M larvae were unsuccessfully developed into adults. Since the authors tested G335M resistance in the larval stage, they should have also included other mutant strains examined in the larval stage to allow for better comparison, at least those with major differences in the adult stages (I276C, I276F+G279S, G279S).

The results from adult bioassays demonstrate greater consistency and reproducibility compared to those from larvae bioassays, offering a more accurate reflection of resistance characteristics. Due to the unavailability of the G335M mutant adult, larvae bioassays were employed as an alternative.

  1. The discussion is incomplete. The authors mentioned different results about I276C from published paper. Moreover, same study successfully generated G335A into the adult stage where G335M failed. The authors should have discussed the differences and potential reasons for the discrepancies between the two studies.

We have revised the discussion as “During the finalization of this manuscript, a separate study produced G335A, I276C and I276F knock-in Drosophila mutants. Interestingly, in their study, homozygosity for I276C was found to be lethal, a result that differs from our findings. Besides, homozygous G335A flies reached adult stage and exhibited a high resistance to broflanilide and isocycloseram, indicating that larger amino acid residues at position 355 impose stronger fitness costs”.

Minor comments:

  1. The citations need improvement. For example:

Line 52-55: Where is the citation about RDL in mosquitoes and pests?

References have been added.

Line 72-74: Citation of the recent study?

References have been added.

Line 276-278: Reference?

References have been added.

Line 278-281: Where do the references support this conclusion?

References have been added.

Round 2

Reviewer 1 Report

Comments and Suggestions for Authors

As mentioned in my reviewer report the scientific advancements are rather incremental. However, the points raised in my review have been addressed.

Comments on the Quality of English Language

Ok